# Mediation of the association between stigma and HIV status and fertility intention by fertility desire among heterosexual couples living with HIV in Kunming, China

Yingwu Guo[1,2], Wit Wichaidit[2], Yingrong Du[1], Jun Liu[1], Virasakdi Chongsuvivatwong[2]*

1 Department of Infectious Diseases, Third People's Hospital of Kunming City, Kunming, People's Republic of China, 2 Faculty of Medicine, Department of Epidemiology, Prince of Songkla University, Hat Yai, Songkhla, Thailand

* cvirasak@medicine.psu.ac.th

**Data Availability Statement:** All relevant data are within the paper and its Supporting Information files.

## Abstract

### Objectives

This study aimed to assess the influences of stigma and HIV status on reproductive intention among heterosexual couples living with HIV in China.

### Methods

A cross-sectional study was conducted in Kunming, China among 315 HIV-positive individuals and their spouses (n = 315 couples). An interview questionnaire was used to collect data on intention, desire, HIV Stigma Scale items, and HIV status. Dyadic fertility intention was examined using the actor-partner interdependence mediation model, based on the Traits-Desires-Intentions-Behavior framework.

### Results

The husbands' or wives' internalized stigma had significant negative effects on their own fertility desire (β = −0.149, p<0.05 and β = −0.238, p<0.01, respectively). HIV-positive status of the husbands was weakly linked to their own fertility intention (β = −0.181, p<0.05). Husbands' perceived provider stigma was associated with their own and their wives' fertility intention via the mediating effect of their fertility desire (β = −0.374, p<0.001 and β = −0.203, p<0.01, respectively). The cumulative influence of their reproductive desire mediated the husband's perceived provider stigma and the wife's internalized stigma on their fertility intention.

### Conclusions

Stigma and HIV status were associated with fertility intention among couples living with HIV, mediated by fertility desire. The high intra-couple correlation suggested that counseling should be conducted when both spouses are present together with extensive discussions

**Funding:** Funding was received from the Higher Education Research Promotion and Thailand's Education Hub for the Southern Region of ASEAN Countries Project Office of the Higher Education Commission (TEH-AC:016/2018). The funders had no role in study design, data collection and analysis, decision to publish, or preparation of the manuscript.

**Competing interests:** The authors declare no conflict of interest.

on concerns regarding HIV-related stigma, potential discrepancies between each partner's fertility desire and intention, and the influence of one partner on the other.

## Introduction

People living with HIV (PLHIV) in a heterosexual relationship face a unique challenge. Although PLHIV may desire to have children similar to their HIV-negative counterparts [1], conception and childbearing in couples with at least one HIV-positive partner may result in horizontal transmission of HIV between partners, or vertical mother-to-child transmission of HIV. Although the likelihood of such transmissions can be reduced through medical interventions, such as fertility clinics and prevention of mother-to-child transmission (PMTCT) services, HIV-related stigmas create barriers in the use of these interventions [2]. Therefore, it is necessary for relevant stakeholders, particularly those who work in reproductive health and HIV prevention and control, to consider the influence of stigma and HIV-positive status on fertility desire and intention among couples living with HIV.

Despite the importance of these considerations, the majority of previous studies were conducted among PLHIV who were single or without the involvement of their partners if in a partnered relationship. Studies in dyads have the potential to increase our understanding of mutual fertility intention [3]. In that regard, the actor-partner interdependence mediation model (APIMeM) is suitable for investigating mutual intention among couples [4,5]. The APIMeM includes two assessments. The first is the possible correlations between the two types of stigmas (i.e., perceived provider and internalized) or HIV status, which would indicate a compositional effect. The second correlation is the residual nonindependence in the variables related to fertility intention, which indicates nonindependence that the APIMeM itself cannot explain.

On a related note, the Traits-Desires-Intentions-Behavior (TDIB) framework [6,7] denotes how the individual's perceived stigma and HIV status (traits) interact with the cognitive notions of desire (want) and intention (plan) to have a child [8,9]. However, there has been no empirical data on the influence of one person on his or her spouse's fertility desire and intention according to this framework. The assessment of mutual fertility intention using the heterosexual couple dyad as the unit of study should greatly improve our understanding of mutual fertility intention.

We hereby hypothesize that stigma and HIV-positive status among couples living with HIV are associated with fertility intention, and mediated by fertility desire, based on the APIMeM (Fig 1) [4,5]. The objectives of this study were to assess the direct influence of stigma and HIV-positive status on fertility intention and the extent that the influence of stigma was mediated by fertility desire among heterosexual HIV-positive husband-wife dyads in Kunming, China.

## Methods

### Study design and setting

A cross-sectional study was conducted at a tertiary hospital in Kunming City, China. Structured interviews were carried out from October to December 2020. Kunming City was chosen for the study because it was the site of an on-going HIV epidemic. Furthermore, the hospital's infectious disease antiretroviral treatment (ART) clinic was one of the pilot sites for a PMTCT program in China, and thus had considerable experience and rapport with the local PLHIV community.

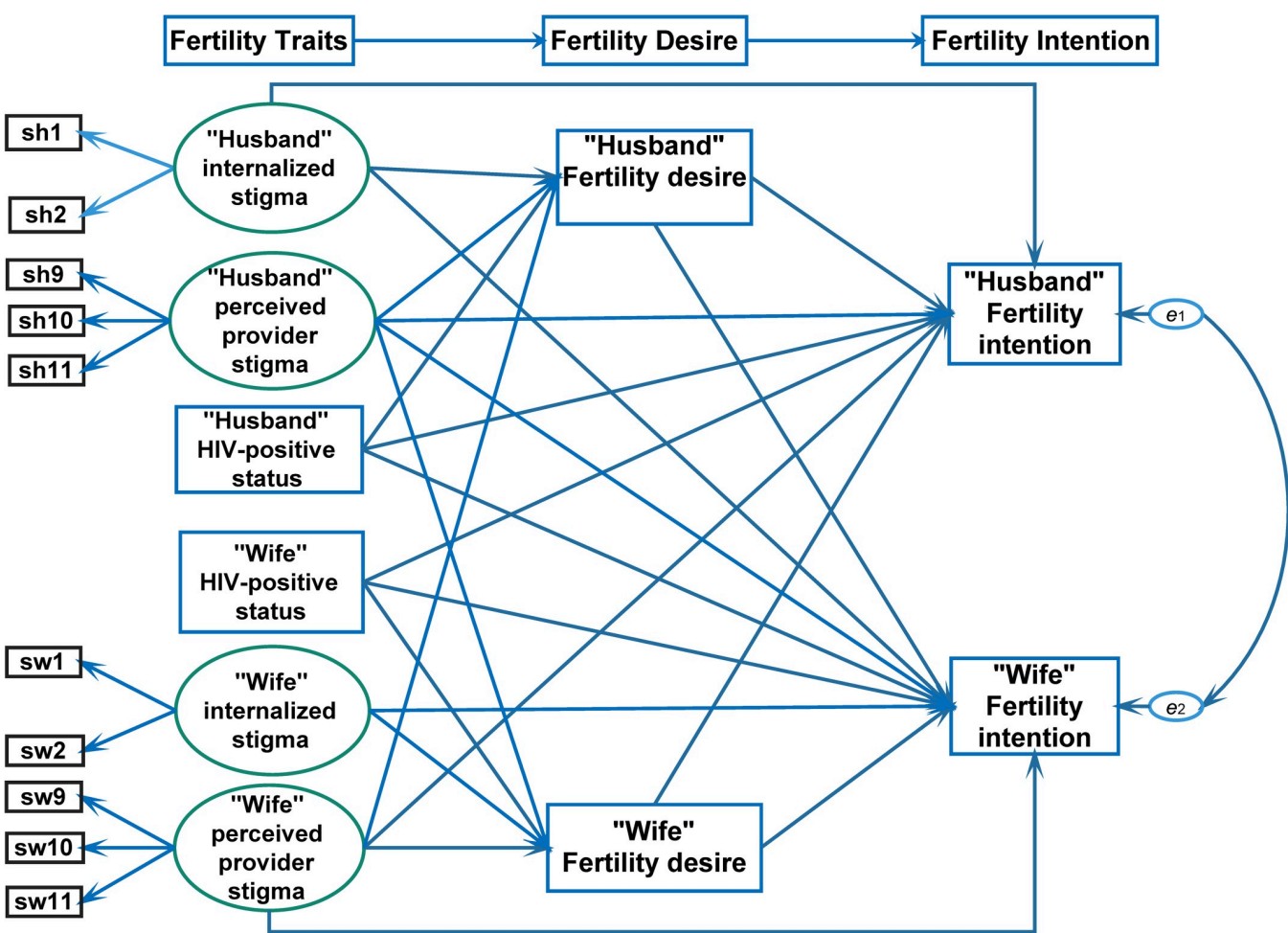

**Fig 1. Hypothesized framework for fertility intention among monogamous HIV concordant and discordant couples.**

### Study participant eligibility criteria and recruitment

The study participants included HIV-infected reproductive-age patients attending an ART clinic at the study hospital. The inclusion criteria included: 1) reproductive age (i.e., 20–40 years for females and 20–50 years for males), 2) stable heterosexual relationship where both partners were aware of the HIV status, 3) no child or 1 child, 4) received ART for at least 12 months with undetectable viral loads, 5) no AIDS-defined illness, and 6) living in Yunnan Province. The exclusion criteria were: 1) unable to communicate, 2) too unwell to participate in the interview, and 3) unable to conceive for any reason (e.g., hysterectomy, oophorectomy, medical illness that contraindicated pregnancy, such as complicated heart disease or active autoimmune disease).

One of the investigators worked at the study hospital and prepared a sampling frame based on the list of current patients at the ART clinic in the ART database. The sampling frame data included the patient's clinic-specific identification number, gender, and the date of the next appointment. The investigators used a computer program to randomly sample the patients stratified by sex. On the day of the appointment, investigators approached each patient in person, introduced themselves, assessed the patient's eligibility, provided information about the study, and inquired about the patient's interest in participation. Each participant was given

approximately 10 minutes to decide whether to participate in the study interview. The investigators then asked for verbal informed consent from the patients before starting the study interview. The patients were also asked for permission to contact their partner to provide information and ask for verbal informed consent. In this study, we refer to one pair of the patient and their partner as a couple.

Initially, a total of 325 couples were invited to participate in the study. Ten couples had at least one spouse who did not complete the consent form. Finally, 315 couples (96.9% of all recruited couples) were included in this analysis.

## Instrument

The study instrument was a structured interview questionnaire. The questionnaire included a section on socio-demographic characteristics (age, educational level, occupation status, ethnic group, registered residence, religion, and medical insurance), HIV status, HIV-related stigma, fertility desire, and fertility intention.

## Measurement of HIV-related stigma

A revised stigma scale was used to measure HIV-related stigma [10–12]. The 5-item scale included two subscales: internalized stigma and stigma from healthcare providers [13,14]. A higher score on the scale indicated a greater level of stigma. The scale is recognized to be valid and reliable by the scientific community. In the current study, the Cronbach's alpha was 0.50 among the couples living with HIV and 0.99 among the spouses, with subscale reliability values ranging from 0.62 to 0.93 among the spouses.

Divergent validity was used to demonstrate that unrelated constructs were actually unrelated by comparing the squared correlations of all latent variables with their average variance extracted values [15].

## Measurement of fertility desire

General desire to have a/another child was assessed using the following question: "Do you want to have a/another child at the present?" The possible responses were "yes", "not sure", or "no" by the couples living with HIV [7].

## Measurement of fertility intention

Fertility intention was assessed with a single question: "How likely are you to have a/another child during the next three years?" The responses were on a five-point scale that ranged from definitely disagree to definitely agree [6].

## Data collection

The data were obtained by four trained research assistants who worked as part of a team from the hospital's ART clinic in Kunming City. After providing information about the study and obtaining verbal informed consent, each respondent was interviewed prior to and separately from his or her spouse. Face-to-face interviews with the patient and their spouse, as well as telephone interviews with the spouse (if living far away), took approximately 15–30 minutes. All questionnaires were returned with no missing information.

## Data analysis

All statistical analyses were performed using R. The socio-economic characteristics and HIV concordant/discordant status were summarized using descriptive statistics. Pearson

correlation coefficients were calculated for the variables measured separately for the wife and husband.

We used the APIMeM to examine the impact of wives' and husbands' reported stigmas on their partner's fertility intention via mediation of their own and their partner's fertility desires [16]. The model examines the influences of the husband (direct and indirect) and the wife (direct and indirect). The spouse effect refers to the extent that an individual's current trait or behavior is influenced by his/her previous trait or behavior and quantifies the trait or behavior's internal consistency or stability. The wife effect refers to the extent that a certain partner affects a woman's trait or behavior and quantifies the association in relational data. This model is divided into two sections and includes ten variables: six independent variables, two outcome variables, and two potential mediator variables. The six independent variables were: 1) wife's internalized stigma, 2) husband's internalized stigma, 3) wife's perceived provider stigma, 4) husband's perceived provider stigma, 5) wife's HIV-positive status, and 6) husband's HIV-positive status. The two outcome variables were: wife's fertility intention and husband's fertility intention. The two potential mediator variables were: wife's fertility desire and husband's fertility desire.

The influences of stigma and HIV status on fertility intention were examined using structural equation modeling. The effects of individual participants on their spouses were estimated and tested using a 5,000 sample size that used the bias-corrected bootstrapping approach with a 95% confidence interval [17].

### Ethical considerations

The Human Research Ethics Committee of Prince of Songkla University (REC-63-208-18-1) and the Third People's Hospital Research Ethics Review Committee authorized this study (2020072001). Pseudonyms were used to protect the participants' identities in this study. Each patient and spouse volunteered and completed an informed consent form. All data were kept secret and confidential, and only the research team had access to the data.

## Results

The socio-economic and HIV status characteristics of the participants are summarized in Table 1. The mean ages and standard deviations of the husbands and wives were 36.82 (5.90) years and 33.81 (5.19) years, respectively. The most common level of educational attainment among the husbands was senior high school, whereas the most common level among the wives was junior high school. Unemployment among the wives was twice as high as among the husbands. Most participants were Han Chinese who lived in rural areas. Most couples (68%) reported the New Rural Cooperative Medical Insurance as their medical insurance provider. The monthly income level varied widely among the study participants. Three-fourths of the couples were sero-discordant with an HIV-positive wife and a HIV-negative husband being slightly more common than vice versa.

### Descriptive statistics and bivariate correlation in the husband-wife dyads

Pearson's correlation analysis (Table 2) showed that the husbands' perceived provider stigma scores were correlated with those of their wives (r = 0.52, p-values <0.05). Significant and positive correlations were found between the husbands and their wives with regards to fertility desire and intention (r = 0.47 and r = 0.73, respectively, all *p*-values <0.01). In this study, all scales had acceptable levels of Cronbach's alpha reliability (range = 0.68–0.98) and average variance extracted divergent validity coefficients (0.44–0.96).

**Table 1. Socio-demographic and HIV-related characteristics.**

| | Husband (n = 315) | Wife (n = 315) | Total (N = 630) |
|---|---|---|---|
| | n (%) | n (%) | N (%) |
| Age category | | | |
| 20–30 | 33 (10.5) | 72 (22.9) | 105 (16.7) |
| 31–35 | 93 (29.5) | 103 (32.7) | 196 (31.1) |
| 36–40 | 93 (29.5) | 101 (32.1) | 194 (30.8) |
| 41+ | 96 (30.5) | 39 (12.4) | 135 (21.4) |
| Education level | | | |
| Primary school | 34 (10.8) | 42 (13.3) | 76 (12.1) |
| Junior school | 94 (29.8) | 123 (39) | 217 (34.4) |
| Senior school | 103 (32.7) | 93 (29.5) | 196 (31.1) |
| Graduate and above | 84 (26.7) | 57 (18.1) | 141 (22.4) |
| Occupation status | | | |
| Government employee | 44 (14) | 18 (5.7) | 62 (9.8) |
| Jobless | 47 (14.9) | 105 (33.3) | 152 (24.1) |
| Manual laborer | 45 (14.3) | 26 (8.3) | 71 (11.3) |
| Private employee | 73 (23.2) | 85 (27) | 158 (25.1) |
| Self-employed | 106 (33.7) | 81 (25.7) | 187 (29.7) |
| Ethnic group | | | |
| Han | 283 (89.8) | 263 (83.5) | 546 (86.7) |
| Others | 32 (10.2) | 52 (16.5) | 84 (13.3) |
| Registered residence | | | |
| Rural | 194 (61.6) | 203 (64.4) | 397 (63) |
| Urban | 121 (38.4) | 112 (35.6) | 233 (37) |
| Religion | | | |
| No | 284 (90.2) | 273 (86.7) | 557 (88.4) |
| Yes | 31 (9.8) | 42 (13.3) | 73 (11.6) |
| Medical insurance status | | | |
| NRCMS | 182 (57.8) | 190 (60.3) | 372 (59) |
| UEBMI | 35 (11.1) | 22 (7) | 57 (9) |
| URBMI | 98 (31.1) | 103 (32.7) | 201 (31.9) |
| **Couple-level variables** | | | |
| Household monthly income (CNY) | | | |
| 0–5,999 | - | - | 90 (28.6) |
| 6,000–9,999 | - | - | 78 (24.8) |
| 10,000–19,999 | - | - | 79 (25.1) |
| 20,000+ | - | - | 68 (21.6) |
| Couple HIV serostatus | | | |
| Both positive | - | - | 84 (26.7) |
| Husband (positive) and wife (negative) | - | - | 103(32.7) |
| Husband (negative) and wife (positive) | - | - | 128 (40.6) |

HIV = human immunodeficiency virus; NRCMS = New Rural Cooperative Medical Insurance Scheme; UEBMI = Urban Employees Basic Medical Insurance; URBMI = Urban Residents Basic Medical Insurance; CNY = Chinese Yuan (Exchange rate: 1 USD = 6.72 CNY).

## Dyadic data analysis using the APIMeM

Fig 2 depicts the estimated paths of direct effects with $R^2$ = 0.49 for husbands and $R^2$ = 0.46 for wives. The model fit the data well: $x^2$/df = 2.59; SRMR = 0.047; RMSEA = 0.071; CFI = 0.97;

**Table 2. Pearson correlation coefficients, mean (standard deviation), and Cronbach's alpha among the study variables (N = 315 couples).**

|  | Mean (SD) | Var 1 | Var 2 | Var 3 | Var 4 | Var 5 | Var 6 | Var 7 | Var 8 | Var 9 | Var 10 |  |
|---|---|---|---|---|---|---|---|---|---|---|---|---|
| Var 1 | 2.4 (1.2) | 1 |  |  |  |  |  |  |  |  |  |  |
| Var 2 | 2.5 (1.2) | −0.17 | 1 |  |  |  |  |  |  |  |  |  |
| Var 3 | 3.1 (0.67) | −0.03 | 0.18* | 1 |  |  |  |  |  |  |  |  |
| Var 4 | 3 (0.72) | 0.04 | 0.52** | -0.1 | 1 |  |  |  |  |  |  |  |
| Var 5 | 0.58(0.49) | -0.25** | -0.24** | -0.04 | -0.17* | 1 |  |  |  |  |  |  |
| Var 6 | 0.62(0.49) | 0.03 | -0.25** | -0.25** | -0.26** | 0.47** | 1 |  |  |  |  |  |
| Var 7 | 3.46(1.13) | -0.13 | -0.34** | -0.15* | -0.21** | 0.57** | 0.53** | 1 |  |  |  |  |
| Var 8 | 3.38(1.23) | -0.02 | -0.28** | -0.18* | -0.26** | 0.51** | 0.65** | 0.73** | 1 |  |  |  |
| Var 9 | 0.59(0.49) | 0.50** | -0.14 | 0.23** | -0.09* | -0.17* | -0.16* | -0.16* | -0.1 | 1 |  |  |
| Var 10 | 0.67(0.47) | -0.34** | 0.16 | -0.45** | 0.21** | 0.06 | 0.08 | 0.08 | 0.04 | -0.58** | 1 |  |
| Cronbach's alpha | - | - | 0.96 | 0.68 | 0.98 | 0.70 | - | - | - | - | - | - |
| Average Variance Extracted |  |  | 0.93 | 0.44 | 0.96 | 0.45 | - | - | - | - | - | - |

Note: Var 1 = Internalized stigma (husband); Var 2 = Perceived provider stigma (husband); Var 3 = Internalized stigma (wife); Var 4 = Perceived provider stigma (wife);
Var 5 = Positive fertility desire (husband); Var 6 = Positive fertility desire (wife); Var 7 = Fertility intention (husband); Var 8 = Fertility intention (wife); Var
9 = Husband's HIV-positive status; Var 10 = Wife's HIV-positive status.

SD = standard deviation

*p<0.05

**p<0.01.

and TLI = 0.95. The covariance of the errors from both sides was statistically significant (β = 0.538, p<0.001), which indicated both husband and wife shared fertility intention. In terms of actor effects, husbands and wives with a high level of fertility desire had a greater intention to have children (β = 0.336, p = 0.000; β = 0.485, p<0.001, respectively). The husband's desire for children had a statistically significant effect on the wife's fertility intention (β = 0.260, p<0.001), which was similar to the counterpart effect from the wife to the husband (β = 0.297, p<0.001).

The pattern of cross-over effects was lacking at the trait level. Traits only directly influenced fertility desire and intention of the same individual and not his or her spouse, which is not shown in Fig 1.

Husbands' or wives' internalized stigma had significant negative effects on their own fertility desire (β = −0.149, p < 0.05 and β = −0.238, p < 0.01, respectively). However, only husbands had a significant negative effect of perceived provider stigma on their own fertility desire (β = −0.226, p < 0.01) and fertility intention (β = −0.234, p < 0.01).

Total effects of stigma and HIV status on fertility intention consists of direct effects and indirect effects. The direct effects were not significant and are not shown in Fig 1. The indirect effects are listed in sections a.1 and a.3 and the total effects are in sections a.2 and b.2 of Table 3.

Husbands' perceived provider stigma was associated with their own and their wives' fertility intention via the mediating total effect of their fertility desire (β = −0.374, p < 0.001 and β = −0.203, p < 0.01, respectively). HIV-positive status of the husbands was weakly linked to their own fertility intention (β = −0.181, p < 0.05). Through the whole mediating effect of their fertility desire, the wives' internalized stigma was linked with their own and their husbands' fertility intention (β = −0.164, p < 0.05 and β = −0.135, p < 0.05, respectively). Both husbands and wives reported that HIV-related reproductive stigma reduced their fertility intention by decreasing their desire to have a/another child, while their fertility intention was related to an increased fertility desire.

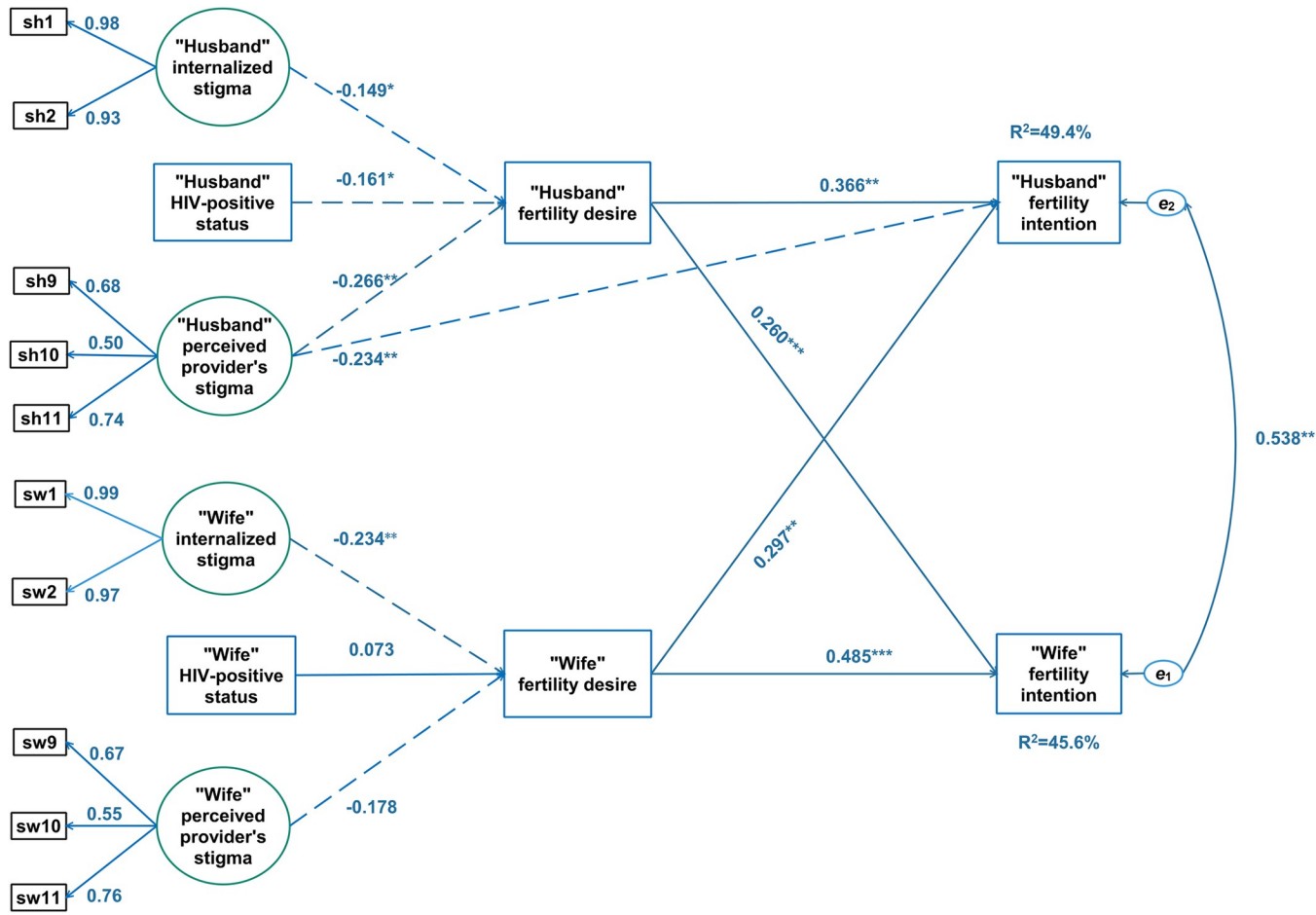

**Fig 2. Associations of wives and husbands' childbearing internalized and provider stigma and fertility intention through fertility desire based on actor-partner interdependence mediation (APIMeM) model.** Significant ($p < .05$) positive effect, significant ($p < .05$) negative effect, ***: $p < .01$, **: $p < .05$; $R^2$: Squared multiple correlation, e1 and e2: Residual error.

## Discussion

We assessed the extent that fertility desire mediated the association between HIV-related stigma and fertility intention among couples living with HIV, as well as the mediation on the association between HIV status and fertility intention. The findings indicated a strong reciprocal relationship within each couple on fertility intentions. Intra-spouse residual correlation of fertility intention was strong. The husbands' fertility intention considerably predicted the wives' fertility intention. Fertility intention of the wives or husbands was strongly influenced by their own fertility intention, their spouses' fertility intention, and their spouses' fertility desire. On the other hand, fertility traits related to HIV status and stigma influenced only one's own fertility desire without significant effects on the spouse. Total effects of trait variables on fertility intention mainly acted through fertility desire. The husbands' perceived provider stigma directly predicted their own fertility intention. Other direct effects were not statistically significant. The novelty of using this analysis for fertility behavior of couples living with HIV would be useful for future studies and health services.

The findings suggested that fertility intention in couples living with HIV occurred as dyads (i.e., there was significant interconnectedness in each couple with regard to fertility intention). The findings were consistent with a previous study, which found that the strongest predictor

**Table 3. Coefficients of indirect and total significant effects in the APIMeM.**

| Path | β | SE | p-value | CI |
|---|---|---|---|---|
| **a. Husband's stigma and HIV positive status→ Fertility desire (husband and wife) → Fertility intention (husband and wife)** | | | | |
| **a.1 Indirect effects** | | | | |
| Provider stigma (Husband) → Fertility intention (husband) | −0.14 | 0.106 | 0.018 | (−0.508, −0.073) |
| Provider stigma (Husband) → Fertility intention (wife) | −0.139 | 0.132 | 0.041 | (−0.568, −0.022) |
| HIV-positive status (husband) → Fertility intention (husband) | −0.096 | 0.103 | 0.03 | (−0.45, −0.032) |
| HIV-positive Status (husband) → Fertility intention (wife) | −0.102 | 0.117 | 0.028 | (−0.52, −0.05) |
| **a.2 Total effects** | | | | |
| Provider stigma (husband) → Fertility intention (husband) | −0.374 | 0.178 | 0.00 | (−1.073, −0.298) |
| Provider stigma (husband) → Fertility intention (wife) | −0.203 | 0.171 | 0.021 | (−0.786, −0.144) |
| HIV-positive status (husband) → Fertility intention (husband) | −0.181 | 0.168 | 0.013 | (−0.732, −0.082) |
| **b. Wife's stigma and HIV-positive status→ Fertility desire (husband and wife) → Fertility intention (Husband and Wife)** | | | | |
| **b.1 Indirect effects** | | | | |
| Internalized stigma (wife) → Fertility intention (wife) | −0.117 | 0.042 | 0.007 | (−0.197, −0.028) |
| **b.2 Total effects** | | | | |
| Internalized stigma (wife) → Fertility intention (wife) | −0.164 | 0.058 | 0.006 | (−0.27, −0.029) |
| Internalized stigma (wife) → Fertility intention (husband) | −0.135 | 0.054 | 0.024 | (−0.229, −0.001) |

Note: β = standardized estimate.

SE = standard error; CI = confidence interval; APIMeM = actor-partner interdependence mediation model.

for a person's intention to have children was having a partner who intended to have children [18]. Other studies also showed that HIV patients reported that their spouse desired to have children but such data were not collected directly from the patient's partners [19–21]. This current study contributed to the body of literature by collecting data from both partners and analyzing the data as dyads, which enabled direct reflection of the couple dyad psyche.

Participants' fertility desire was found to be strongly associated with their own fertility intention, which was similar to the findings of a previous study [6]. Furthermore, there was mutual influence of the participants' own fertility desire on the fertility intention of their spouses. A plausible explanation for these results was that among HIV-positive individuals a couple's relationship status was a significant predictor of the partner's fertility desire [14]. The support and encouragement of a spouse with a positive desire for children may be a significant factor in fertility intentions [7,22]. Furthermore, fertility desire is the most important mediating factor for fertility intention among HIV patients, and it is a key variation that has an important impact on the active response to fertility behavior [7].

No significant association was observed between demographic-socio-economic characteristics and fertility intention. Previous studies, on the other hand, showed a close link between gender and perceived stigma from healthcare providers [8,23,24]. These disparities by gender also could have accounted for the stronger association among the husbands compared to the wives.

Wives' internalized stigma was more strongly associated with the husbands' fertility intention via fertility desire than vice versa. It was possible that HIV-positive women were more

sensitive to internalized stigma than their husbands. Some of the discrepancies between this study's findings and those of earlier studies might be explained by the cultural context of this study. In Chinese society and culture, wives are primarily reliant on their husbands for fertility [25–27]. Furthermore, it seems that the wives' fertility intention was more influenced by the efforts of their husbands to address stigma, fertility desire, and fertility intention [24,28] than from other factors.

Only minor differences were found in fertility intention between HIV-positive versus HIV-negative husbands. However, the differences were more noticeable among HIV-positive versus HIV-negative wives. Consistent with previous research in Mozambique [29], no association was found between a woman's HIV status and fertility desire. HIV-positive women were found to be less likely to have children than HIV-negative women [30]. These contradictory findings could be attributed in part to the widespread belief that childbirth was incompatible with HIV. However, PMTCT programs have been available in the study area for more than 20 years, which might have reduced the fear of HIV vertical transmission, particularly among the women [31].

## Strengths and limitations

Our study had a fairly high participation rate from the recruited couples, which potentially reduced the possibility of selection bias. However, a number of limitations should be considered in the interpretation of our study findings. First, the causal pathway in this article was based on an existing theory with a relatively straightforward pathway and a focus on the couple dyad. Findings from a more inductive or inferential approach could have yielded different results. Second, the level of stigma experienced by each participant might have varied over time but our cross-sectional study design precluded the measurement of these variations and reflected mainly the self-reported stigma at the time of the interview. Findings from a longitudinal study may yield different results. Third, this study was conducted in Yunnan Province, which is an area in China with high ethnic diversity and a relatively high level of poverty. The findings of this study may have limited generalizability to couples living with HIV in other settings.

## Conclusions

We assessed the extent that fertility desire mediated the association between HIV-related stigma and fertility intention among couples living with HIV, as well as the mediation on the association between HIV status and fertility intention. The outcomes of this study show the importance of stigma in intra-couple fertility intention. The study findings have implications for family planning counseling in couples living with HIV. The high intra-couple correlation between fertility desire and intention suggests that couples should receive counseling on fertility together, with extensive discussions on concerns regarding HIV-related stigma, potential discrepancies between each partner's fertility desire and intention, and the influence of one partner on the other.

## Supporting information

**S1 Appendix.**
(TIF)

**S1 File.**
(CSV)

## Acknowledgments

This study is a part of the first author's thesis in partial fulfillment of the requirements for a Ph. D. degree in Epidemiology at Prince of Songkla University, Songkhla, Thailand. We greatly appreciate the assistance from the staff members of the HIV treatment center who supported our study at the Third People's Hospital in Kunming, China.

## Author Contributions

**Conceptualization:** Yingwu Guo, Wit Wichaidit, Virasakdi Chongsuvivatwong.

**Data curation:** Yingwu Guo, Yingrong Du.

**Formal analysis:** Yingwu Guo, Wit Wichaidit, Jun Liu, Virasakdi Chongsuvivatwong.

**Funding acquisition:** Yingwu Guo.

**Investigation:** Yingwu Guo, Yingrong Du, Jun Liu.

**Methodology:** Yingwu Guo, Wit Wichaidit, Virasakdi Chongsuvivatwong.

**Project administration:** Yingwu Guo, Yingrong Du, Virasakdi Chongsuvivatwong.

**Resources:** Yingrong Du, Jun Liu.

**Supervision:** Virasakdi Chongsuvivatwong.

**Validation:** Yingwu Guo, Virasakdi Chongsuvivatwong.

**Writing – original draft:** Yingwu Guo.

**Writing – review & editing:** Wit Wichaidit, Virasakdi Chongsuvivatwong.

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
