## [Decision Letter · Decision Letter 0]

14 Oct 2022

PONE-D-22-21531Mediation of the association between stigma and HIV status and fertility intention by fertility desire among heterosexual couples living with HIV in Kunming, ChinaPLOS ONE

Dear Dr Chongsuvivatwong,

Thank you for submitting your manuscript to PLOS ONE. After careful consideration, we feel that it has merit but does not fully meet PLOS ONE’s publication criteria as it currently stands. Therefore, we invite you to submit a revised version of the manuscript that addresses the points raised during the review process.

We look forward to receiving your revised manuscript.

Kind regards,

Deepak Dhamnetiya, MD

Academic Editor

PLOS ONE

“Funding was received from the Higher Education Research Promotion and Thailand’s Education Hub for the Southern Region of ASEAN Countries Project Office of the Higher Education Commission (TEH-AC:016/2018)”

Reviewers' comments:

Reviewer's Responses to Questions

**Comments to the Author**

1. Is the manuscript technically sound, and do the data support the conclusions?

Reviewer #1: Yes

Reviewer #2: Yes

Reviewer #3: Yes

2. Has the statistical analysis been performed appropriately and rigorously? 

Reviewer #1: Yes

Reviewer #2: Yes

Reviewer #3: Yes

3. Have the authors made all data underlying the findings in their manuscript fully available?

Reviewer #1: Yes

Reviewer #2: Yes

Reviewer #3: Yes

4. Is the manuscript presented in an intelligible fashion and written in standard English?

Reviewer #1: Yes

Reviewer #2: Yes

Reviewer #3: Yes

5. Review Comments to the Author

Reviewer #1: The manuscript is well written. There are a few minor suggestions:

i) Overall, there are typos and grammatical errors detected. A professional proof reading is advised.

ii) In the methods section the sampling strategy needs to be more clearly described and also add in more details about the study tool (validity etc.)

iii) In the discussion section study limitations and justifications need to be explicitly outlined.

Reviewer #2: 1.Introduction should be written more clearly and in simple terms rather than just quoting other studies

2. Reframe the objectives- should be more clear

3. When using abbreviations, make sure to expand it when using it for the first time in the manuscript (Ex APIMeM)

4. Overall Language of the manuscript should be more clear, precise

5. Key to reading figure should accompany the figure rather than in the middle of the manuscript

6. What was the amount of time given for giving consent after explaining to the study participants?

7. What are the other recommendations from the study?

Reviewer #3: The manuscript is well written, technically sound and the conclusions were drawn from the results following a robust statistical analysis. The limitations outline further strengthens the work as it clearly aligns the findings with what the study aimed to achieve. I opined that the manuscript needs no further modifications.

6. PLOS authors have the option to publish the peer review history of their article (what does this mean?). If published, this will include your full peer review and any attached files.

Reviewer #1: No

Reviewer #2: No

Reviewer #3: **Yes: **Mohammed Abdullahi Abdulkarim

---

## [Author Response · Author response to Decision Letter 0]

28 Oct 2022

Dear Editor and Reviewers,

We have responded to your comments in a separate file and submitted along with the revised manuscript. If you need anything else, please let us know.

Thank you very much

Professor Virasakdi Chongsuvivatwong

---

## [Decision Letter · Decision Letter 1]

14 Nov 2022

Mediation of the association between stigma and HIV status and fertility intention by fertility desire among heterosexual couples living with HIV in Kunming, China

PONE-D-22-21531R1

Dear Dr. Chongsuvivatwong,

We’re pleased to inform you that your manuscript has been judged scientifically suitable for publication and will be formally accepted for publication once it meets all outstanding technical requirements.

Kind regards,

Deepak Dhamnetiya, MD

Academic Editor

PLOS ONE

Additional Editor Comments (optional):

Reviewers' comments:

Reviewer's Responses to Questions

**Comments to the Author**

1. If the authors have adequately addressed your comments raised in a previous round of review and you feel that this manuscript is now acceptable for publication, you may indicate that here to bypass the “Comments to the Author” section, enter your conflict of interest statement in the “Confidential to Editor” section, and submit your "Accept" recommendation.

Reviewer #2: All comments have been addressed

2. Is the manuscript technically sound, and do the data support the conclusions?

Reviewer #2: Yes

3. Has the statistical analysis been performed appropriately and rigorously? 

Reviewer #2: Yes

4. Have the authors made all data underlying the findings in their manuscript fully available?

Reviewer #2: Yes

5. Is the manuscript presented in an intelligible fashion and written in standard English?

Reviewer #2: Yes

6. Review Comments to the Author

Reviewer #2: All the raised comments have been addressed by the authors. The article can be accepted for publication from my side.

7. PLOS authors have the option to publish the peer review history of their article (what does this mean?). If published, this will include your full peer review and any attached files.

Reviewer #2: No

---

## [Editor Report · Acceptance letter]

24 Nov 2022

PONE-D-22-21531R1 

Mediation of the association between stigma and HIV status and fertility intention by fertility desire among heterosexual couples living with HIV in Kunming, China 

Dear Dr. Chongsuvivatwong:

I'm pleased to inform you that your manuscript has been deemed suitable for publication in PLOS ONE. Congratulations! Your manuscript is now with our production department. 

Kind regards, 

on behalf of

Dr. Deepak Dhamnetiya 

Academic Editor

PLOS ONE